# PDBrenum: A webserver and program providing Protein Data Bank files renumbered according to their UniProt sequences

**Bulat Faezov[1,2], Roland L. Dunbrack, Jr.** [2]*

**1** Institute of Fundamental Medicine and Biology, Kazan Federal University, Kazan, Russian Federation, **2** Institute for Cancer Research, Fox Chase Cancer Center, Philadelphia, Pennsylvania, United States of America

* roland.dunbrack@fccc.edu

**Data Availability Statement:** The data used in this study are derived from data on PDBrenum (http://dunbrack3.fccc.edu/PDBrenum) and can be

## Abstract

The Protein Data Bank (PDB) was established at Brookhaven National Laboratories in 1971 as an archive for biological macromolecular crystal structures. In mid 2021, the database has almost 180,000 structures solved by X-ray crystallography, nuclear magnetic resonance, cryo-electron microscopy, and other methods. Many proteins have been studied under different conditions, including binding partners such as ligands, nucleic acids, or other proteins; mutations, and post-translational modifications, thus enabling extensive comparative structure-function studies. However, these studies are made more difficult because authors are allowed by the PDB to number the amino acids in each protein sequence in any manner they wish. This results in the same protein being numbered differently in the available PDB entries. For instance, some authors may include N-terminal signal peptides or the N-terminal methionine in the sequence numbering and others may not. In addition to the coordinates, there are many fields that contain structural and functional information regarding specific residues numbered according to the author. Here we provide a webserver and Python3 application that fixes the PDB sequence numbering problem by replacing the author numbering with numbering derived from the corresponding UniProt sequences. We obtain this correspondence from the SIFTS database from PDBe. The server and program can take a list of PDB entries or a list of UniProt identifiers (e.g., "P04637" or "P53_HUMAN") and provide renumbered files in mmCIF format and the legacy PDB format for both asymmetric unit files and biological assembly files provided by PDBe.

## Introduction

The Protein Data Bank (PDB) is a database for the three-dimensional structural data of biological macromolecules, including proteins and nucleic acids [1]. The data, typically obtained by X-ray crystallography, NMR spectroscopy, or cryo-electron microscopy, and submitted by scientists from around the world, are freely accessible through the World Wide PDB (wwPDB) http://www.wwpdb.org/ and three wwPDB partner sites, https://www.rcsb.org [2], https://

accessed following the protocol outlined in the Methods section.

**Funding:** This work was funded by National Institutes of Health grant R35 GM122517 (R.L.D.), https://www.nigms.nih.gov/. The funders had no role in study design, data collection and analysis, decision to publish, or preparation of the manuscript.

**Competing interests:** The authors have declared that no competing interests exist.

www.ebi.ac.uk/pdbe [3], and https://pdbj.org [4]. The PDB provides useful and fundamental information about tens of thousands of proteins. For many proteins, there are 10s or even 100s of available structures performed under varying conditions, including the presence of different binding partners such as inhibitors, nucleic acids, or other proteins, or with mutations and post-translational modifications. However, in each structure in the PDB, authors are allowed to number protein sequences in any way they wish. This includes the coordinates and any functional or structural annotations contained within the PDB files. Authors commonly number according to sequences deposited in gene databanks such as GenBank [5] or UniProt [6]. So, for instance, a domain from a protein that is not at the N-terminus of the natural sequence may start with its position in the full-length sequence. However, different authors may choose different conventions for this numbering. Authors may or may not include the N-terminal methionine or N-terminal signal sequences in the numbering, both of which may be cleaved off to form the mature protein. For example, in PDB entry 3lvp [7], which is a structure of the kinase domain of human IGF1R, the DFG motif amino acids are numbered 1153–1155. But in PDB entry 3d94 [8], these residues are labeled as residues 1123–1125, because the numbering is that of the mature protein, which does not include the 30 amino acid signal sequence cleaved from the N-terminus of the preprotein. Proteins in the PDB often include N-terminal sequence tags, and the numbering of these residues can be just about anything including negative numbers, 0, or numbers that seem to indicate the residues are from the same gene as the protein under study. These inconsistent numbering schemes compromise structural bioinformatics studies that seek to compare multiple structures of a single protein or structures within protein families across the PDB. They also affect mapping of sequence annotation data (such as mutation data) to structural information in the PDB, since any structure downloaded from the PDB may or may not have the same numbering scheme as the sequence database.

The problem of inconsistent numbering, insertion codes, negative residue numbers, and other problems have been discussed previously but not addressed in any rigorous way (e.g. https://proteopedia.org/wiki/index.php/Unusual_sequence_numbering and https://www3.cmbi.umcn.nl/wiki/index.php/Residue_number). Mapping PDB structures to UniProt has been attempted a number of times, including SSMAP [9], Seq2Struct [10], and PDBSWS [11], although these servers did not renumber actual coordinate files, instead only providing the mapping of residue numbers from the PDB to UniProt. Only the PDBSWS server is still functioning.

In this paper, we present downloadable computer code in Python3 and a webserver that provide PDB files where the amino acids in all fields are renumbered according to their UniProt sequences. We obtain the correspondence between protein sequences in PDB chains with UniProt entries from the SIFTS database available from PDBe (https://www.ebi.ac.uk/pdbe/docs/sifts/) [12]. Our program and webserver provide renumbered files in mmCIF [13] and legacy PDB format [14] for both asymmetric unit files (the coordinates deposited by authors) obtained from the RCSB server and biological assembly files (provided by the authors or calculated with the program PISA [15]) in mmCIF format obtained from PDBe, which distributes them for all PDB entries. The webserver is easy to use—the user enters a list of PDB codes ("1abc") or UniProt identifiers in the accession code ("P04637") or SwissProt ID ("P53_HUMAN") format, selects which kinds of files to download (mmCIF and/or legacy-PDB format; asymmetric units and/or biological assemblies), and with one click enables the download of a zip file containing the requested files.

## Methods

PDBrenum was written in Python with use of Python 3.6, BioPython 1.76 [16], Pandas 0.25.1 [17], and Numpy 1.17 modules within Jupyter-Notebook 6.0.1 [18] and PyCharm 2020.2

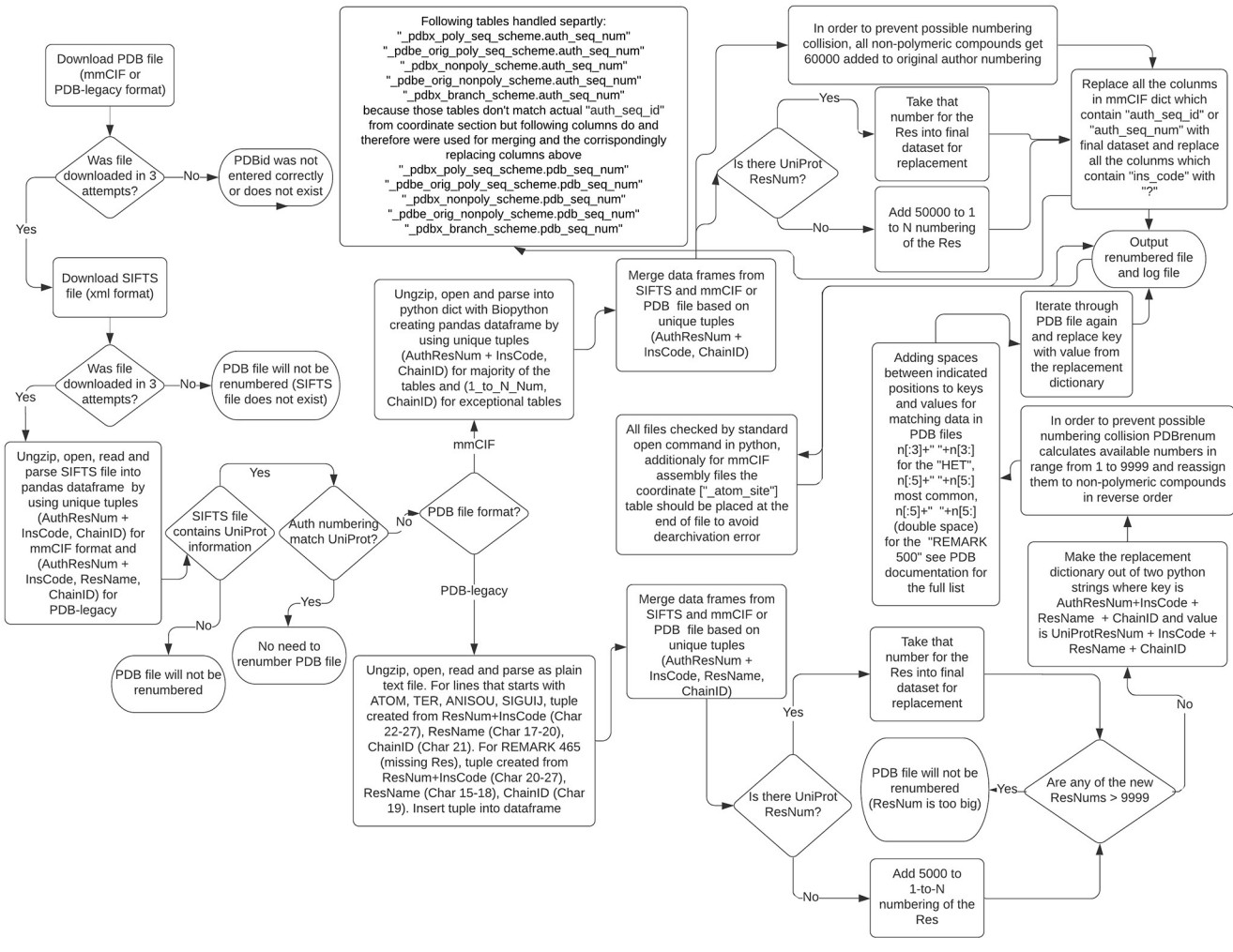

**Fig 1. Flow-chart describing basic procedure of PDBrenum.**

(https://www.jetbrains.com/pycharm/) as an integrated development environment on a Ubuntu 20.04 operating system.

Fig 1 represents a basic scheme of the PDBrenum workflow. First, PDBrenum downloads the structure files (in PDB or/and mmCIF format) and corresponding SIFTS files (in.xml format). The program downloads files in three attempts; if there is no success in three attempts, the assumption is that there is no such file (sometimes servers might not respond or respond with errors, but it is very unlikely to get three bad responses from the server in a row). PDBrenum then parses the SIFTS file to obtain numbering data for each amino acid in each protein chain in the file and places the results in a Pandas dataframe with the following data fields:

- **PDBChainID**: the `label_asym_id` in mmCIF coordinates (from `entityId` in SIFTS). Note: this does not correspond to `entity_id` in the mmCIF files, which instead is an integer that indicates the molecule identity (i.e., each protein sequence and each ligand type gets an `entity_id`).

- **AuthChainID**: the `auth_asym_id` in mmCIF coordinates (from `PDB:dbChainID` in SIFTS).

- *SeqResNum*: the `label_seq_id` in mmCIF coordinates, which is the number of each residue in each protein construct when numbered from 1 to N, the number of residues in the protein chain (from `PDBe:dbResNum` in the SIFTS file).

- *AuthResNum*: the `auth_seq_id` in mmCIF coordinates, which is the author residue number (from `PDB:dbResNum` in SIFTS).

- *InsCode*: the `pdbx_PDB_ins_code` labels in mmCIF coordinates, which are insertion codes, if any, attached to residue numbers in legacy PDB files to distinguish residues inserted in the sequence (from upper-case letters in `PDB:dbResNum` in SIFTS).

- *ResName*: the `label_comp_id` and `auth_comp_id` in mmCIF coordinates, which is the residue name in three-letter code (from `PDB:dbResName` in SIFTS).

- *AccessionID*: for UniProt entry (if any) (from `UniProt:dbAccessionId` in SIFTS).

- *UniProtResNum*: residue number in UniProt reference sequence (if any) (from `UniProt:dbResNum` in SIFTS).

- *UniProtResName*: amino acid type in UniProt sequence (if any) in one-letter code (from `UniProt:dbResName` in SIFTS).

The typical Pandas dataframe will look like the ones shown in Fig 2. The dataframe correlates different numbering systems for the amino acids in each protein chain. The unique residue numbering is the 1 to N numbering (SeqResNum) for each chain, where N is the chain length. This is referred to as `label_seq_id` in the mmCIF file coordinate (`_atom_site`) records. The tuple representing (SeqResNum, ResName, and PDBChainID), denoted `label_seq_id`, `label_comp_id`, and `label_asym_id` in the mmCIF coordinates, is shown in the first column where the combination of the residue number and chain id act as a Pandas index or key for the table. The second numbering system is that used by the authors in the coordinates of the mmCIF file, which is represented in the "PDB" column. It consists of tuples (AuthResNum + InsCode, ResName, and AuthChainID). These values are denoted `auth_seq_id`, `pdbx_PDB_ins_code`, `auth_comp_id`, and `auth_asym_id` in the coordinate section of mmCIF files respectively. Insertion codes are letters attached to some residue numbers by authors to create new residue identifiers for inserted residues in a sequence. They are common in antibody numbering systems [19].

For most amino acids in the PDB, the SIFTS database has a UniProt reference and residue number, which is given in the 3rd column in each dataframe. When there is no UniProt number given in SIFTS for some residues in a chain (usually for sequence tags), we place the number 50,000 in this column. The resulting numbering system (given in the column labeled "UniProt_50k") that PDBrenum will use as a replacement for the author numbering system is the UniProt number where it is available and 50,000+SeqResNum when there is no UniProt number. This guarantees that there will be no collision between a UniProt residue number, and the numbers assigned to sequence tags and other insertions that are not part of the UniProt numbering system.

After reading and processing the SIFTS file for an entry, PDBrenum uncompresses and reads the gzipped mmCIF file as a Python dictionary using BioPython. The dictionary created by the BioPython function MMCIF2DICT forms keys from each mmCIF table and item name as a single string (e.g., `_atom_site.Cartn_x`). The corresponding value for each key is a Python list (e.g., the x-coordinates for all atoms in an entry).

(A)

| PDBe | PDB | UniProt | AccessionID | Uni_or_50k |
|---|---|---|---|---|
| (1, ARG, A) | (null, ARG, A) | 50000 | NaN | 50001 |
| (2, GLY, A) | (null, GLY, A) | 50000 | NaN | 50002 |
| (3, SER, A) | (null, SER, A) | 50000 | NaN | 50003 |
| (4, HIS, A) | (null, HIS, A) | 50000 | NaN | 50004 |
| (5, HIS, A) | (null, HIS, A) | 50000 | NaN | 50005 |
| (6, HIS, A) | (null, HIS, A) | 50000 | NaN | 50006 |
| (7, HIS, A) | (null, HIS, A) | 50000 | NaN | 50007 |
| (8, HIS, A) | (null, HIS, A) | 50000 | NaN | 50008 |
| (9, HIS, A) | (null, HIS, A) | 50000 | NaN | 50009 |
| (10, GLY, A) | (null, GLY, A) | 50000 | NaN | 50010 |
| (11, SER, A) | (0, SER, A) | 50000 | NaN | 50011 |
| (12, ALA, A) | (1, ALA, A) | (54, A, A) | P30044 | 54 |
| (13, PRO, A) | (2, PRO, A) | (55, P, A) | P30044 | 55 |
| (14, ILE, A) | (3, ILE, A) | (56, I, A) | P30044 | 56 |
| (15, LYS, A) | (4, LYS, A) | (57, K, A) | P30044 | 57 |
| (16, VAL, A) | (5, VAL, A) | (58, V, A) | P30044 | 58 |
| (17, GLY, A) | (6, GLY, A) | (59, G, A) | P30044 | 59 |
| (18, ASP, A) | (7, ASP, A) | (60, D, A) | P30044 | 60 |
| (19, ALA, A) | (8, ALA, A) | (61, A, A) | P30044 | 61 |

(B)

| PDBe | PDB | UniProt | AccessionID | Uni_or_50k |
|---|---|---|---|---|
| (1, HIS, A) | (-2, HIS, A) | 50000 | NaN | 50001 |
| (2, HIS, A) | (-1, HIS, A) | 50000 | NaN | 50002 |
| (3, MET, A) | (1, MET, A) | (1, M, A) | P21856 | 1 |
| (4, ASP, A) | (2, ASP, A) | (2, D, A) | P21856 | 2 |
| (5, GLU, A) | (3, GLU, A) | (3, E, A) | P21856 | 3 |
| (6, GLU, A) | (4, GLU, A) | (4, E, A) | P21856 | 4 |
| (7, TYR, A) | (5, TYR, A) | (5, Y, A) | P21856 | 5 |
| (8, ASP, A) | (6, ASP, A) | (6, D, A) | P21856 | 6 |

(C)

| PDBe | PDB | UniProt | AccessionID | Uni_or_50k |
|---|---|---|---|---|
| (81, PHE, A) | (100, PHE, A) | (82, F, A) | Q4PRK9 | 82 |
| (82, THR, A) | (101, THR, A) | (83, T, A) | Q4PRK9 | 83 |
| (83, LYS, A) | (102, LYS, A) | (84, K, A) | Q4PRK9 | 84 |
| (84, ALA, A) | (103A, ALA, A) | (85, A, A) | Q4PRK9 | 85 |
| (85, PRO, A) | (103B, PRO, A) | (86, P, A) | Q4PRK9 | 86 |
| (86, GLY, A) | (103C, GLY, A) | (87, G, A) | Q4PRK9 | 87 |
| (87, LYS, A) | (103D, LYS, A) | (88, K, A) | Q4PRK9 | 88 |
| (88, SER, A) | (103E, SER, A) | (89, S, A) | Q4PRK9 | 89 |
| (89, ASP, A) | (105A, ASP, A) | (90, D, A) | Q4PRK9 | 90 |
| (90, LYS, A) | (105B, LYS, A) | (91, K, A) | Q4PRK9 | 91 |

**Fig 2. Small fragments of the Pandas dataframes assembled from SIFTS files: (A) 2vl3, (B) 2aa3, and (C) 1d5t.** Entry 2vl3 contains a His tag that is not observed in the coordinates. Chain D of 2aa3 contains insertion codes (column 2) for some residues. Entry 1d5t contains a His tag with negative author residue numbers (column 2). The PDBe column in each image contains data for each amino acid in tuples (SeqResNum, ResName, and EntityId), where SeqResNum is the position of the amino acid in the sequence numbered from 1 to N (the length of the sequence). This field acts as the Pandas dataframe index for the whole table, since it is unique for each amino acid. The PDB column contains tuples (AuthResNum + InsCode, ResName, and ChainID). The UniProt column contain tuples of (UniProtResNum, UniProtResName, ChainID) and if there is no UniProt residue number, it contains the number "50,000". The next column contains the UniProt AccessionID. The column UniProt_50k provides the final numbering of residues in the PDBrenum output file: it is the UniProt number when it is available, and 50,000+SeqResNum when there is no UniProt for a chain that has a UniProt. Chains with no UniProt in SIFTS are not renumbered.

In order to renumber PDB files according to UniProt from SIFTS, we need to identify corresponding values in all tables in the mmCIF files and all records in the PDB format files. SIFTS contains the PDBChainIDs, the author ChainIDs, the author residue numbers (with appended insert codes, if they exist), and the 1-to-N numbering of each chain. For each table (e.g. `_atom_site` or `_pdbx_validate_torsion`), we detect whether the table has residue numbers and chain identifiers that may be compared to the SIFTS data described above.

For the ChainIDs, tables may contain both the author ChainIDs and the PDB's ChainIDs, but in some tables only the author ChainIDs exist (e.g., table `_struct_ref_seq`) and are labeled either as `auth_asym_id`, `pdb_strand_id` or `pdbx_strand_id`. These ChainIDs agree with the auth_asym_id in the coordinates. According to the mmCIF dictionary, all variants (with suffixes and prefixes (e.g., `struct_sheet_range.beg_auth_asym_id`) of `auth_asym_id`, `pdb_strand_id`, and `pdbx_strand_id`) correspond to the author ChainIDs in the coordinates and SIFTS, and thus can be used by PDBrenum to translate protein residue numbering into UniProt.

Many tables do *not* contain the 1-to-N numbering of each chain (SeqResNum in Fig 2), and so we need to interpret the values in the author numbering (with insert codes, if any) in each table in order to renumber according to UniProt from SIFTS. Author sequence numbering may be designated as `auth_seq_id`, and may be prefixed or suffixed with other identifiers,

e.g., `auth_seq_id_1` or `beg_auth_seq_id`. We used the mmCIF dictionary to determine that all identifiers that contain `auth_seq_id` or its variants are children of `_atom_site.auth_seq_id` in the coordinate records, i.e. the author residue numbering in the coordinates that corresponds to the author numbering in SIFTS.

mmCIF files contain three tables that provide information on the sequence numbering of molecules in the structure: `_pdbx_poly_seq_scheme`, `_pdbx_nonpoly_seq_scheme`, and `_pdbx_branch_scheme`. These tables include four residue numbering schemes: `seq_id`, `pdb_seq_num`, `ndb_seq_num`, and auth_seq_num. For proteins and other polymers, `seq_id` in the `_pdbx_poly_seq_scheme` is the 1 to N numbering of the chain. These numbers are also found in the `ndb_seq_num` column. The `pdb_seq_num` column corresponds to the residue numbering in the coordinates, according to the mmCIF dictionary. The dictionary indicates that the `auth_seq_num` in the three tables may or may not correspond to the numbering in the coordinates. We found about 2000 files where these numbers differ from `pdb_seq_num`. These residue numbers appear to be those originally deposited by the authors and in these 2000 files, the "author" residue numbering has been altered by the PDB in the coordinates and other tables, and in the legacy PDB format files. This information is apparently kept in the file for reference. It is not used in any other table in any current PDB entry or in the legacy PDB format files. As it turns out, there is only one instance of a similar prefixed identifier, `pdbx_auth_seq_num` (`_struct_ref_seq_dif.pdbx_auth_seq_num`), and it corresponds to `pdb_seq_num`, not `auth_seq_num` in the `pdbx_poly_seq_scheme` table, so it is renumbered by PDBrenum according to SIFTS.

Insert codes may be designated `ins_code`, `PDB_ins_code`, or `pdb_ins_code`, and these names may contain prefixes and suffixes. According to the dictionary, all of these are children of `_atom_site.pdbx_PDB_ins_code`, and thus will agree with any insert codes present in SIFTS.

PDBrenum forms a Pandas dataframe for each mmCIF table (e.g. `_atom_site`) with index names equal to the Python tuple consisting of (AuthResNum + InsCode, ChainID), and merges it with the same combination (labeled "PDB" in Fig 2) from the SIFTS dataframe. This merged table is then used to replace the AuthResNum values with the UniProtResNum or 50,000+SeqResNum values. Non-polymeric molecule types such as small ligands are also renumbered as 60,000+their residue number (`_pdbx_nonpoly_seq_scheme.pdb_seq_num`) in the mmCIF file. With a tuple consisting of the author chainID the author residue number, and the insert code (if any), the following items in mmCIF files are renumbered:

```
_atom_site_anisotrop.pdbx_auth_seq_id
_atom_site.auth_seq_id
_pdbx_distant_solvent_atoms.auth_seq_id
_pdbx_refine_tls_group.beg_auth_seq_id
_pdbx_refine_tls_group.end_auth_seq_id
_pdbx_struct_chem_comp_diagnostics.auth_seq_id
_pdbx_struct_conn_angle.ptnr1_auth_seq_id
_pdbx_struct_conn_angle.ptnr2_auth_seq_id
_pdbx_struct_conn_angle.ptnr3_auth_seq_id
_pdbx_struct_mod_residue.auth_seq_id
_pdbx_struct_sheet_hbond.range_1_auth_seq_id
_pdbx_struct_sheet_hbond.range_2_auth_seq_id
_pdbx_struct_special_symmetry.auth_seq_id
_pdbx_unobs_or_zero_occ_atoms.auth_seq_id
_pdbx_unobs_or_zero_occ_residues.auth_seq_id
_pdbx_validate_chiral.auth_seq_id
```

```
_pdbx_validate_close_contact.auth_seq_id_1
_pdbx_validate_close_contact.auth_seq_id_2
_pdbx_validate_main_chain_plane.auth_seq_id
_pdbx_validate_peptide_omega.auth_seq_id_1
_pdbx_validate_peptide_omega.auth_seq_id_2
_pdbx_validate_planes.auth_seq_id
_pdbx_validate_polymer_linkage.auth_seq_id_1
_pdbx_validate_polymer_linkage.auth_seq_id_2
_pdbx_validate_rmsd_angle.auth_seq_id_1
_pdbx_validate_rmsd_angle.auth_seq_id_2
_pdbx_validate_rmsd_angle.auth_seq_id_3
_pdbx_validate_rmsd_bond.auth_seq_id_1
_pdbx_validate_rmsd_bond.auth_seq_id_2
_pdbx_validate_symm_contact.auth_seq_id_1
_pdbx_validate_symm_contact.auth_seq_id_2
_pdbx_validate_torsion.auth_seq_id
_struct_conf.beg_auth_seq_id
_struct_conf.end_auth_seq_id
_struct_conn.ptnr1_auth_seq_id
_struct_conn.ptnr2_auth_seq_id
_struct_mon_prot_cis.auth_seq_id
_struct_mon_prot_cis.pdbx_auth_seq_id_2
_struct_ncs_dom_lim.beg_auth_seq_id
_struct_ncs_dom_lim.end_auth_seq_id
_struct_sheet_range.beg_auth_seq_id
_struct_sheet_range.end_auth_seq_id
_struct_site_gen.auth_seq_id
_struct_site.pdbx_auth_seq_id
_pdbx_nonpoly_scheme.pdb_seq_num
_pdbx_poly_seq_scheme.pdb_seq_num
_pdbx_branch_scheme.pdb_seq_num
_struct_ref_seq_dif.pdbx_auth_seq_num
_struct_ref_seq.pdbx_auth_seq_align_beg
_struct_ref_seq.pdbx_auth_seq_align_end
```

As noted above, the *pdbx_poly_seq_scheme*, *pdbx_nonpoly_seq_scheme*, and the *pdbx_branch_scheme* (for branched sugars) contain the 1-to-N numbering (called *seq_id*), the author residue numbering corresponding to the coordinates (*pdb_seq_num*), and an extra residue number (*auth_seq_num*) that may differ from *pdb_seq_num*, containing historical data from the original file deposition. When it differs from *pdb_seq_num*, it is not used elsewhere in the mmCIF files. We renumber *pdb_seq_num* according to UniProt, and replace *auth_seq_num* in this table with the values of *pdb_seq_num* in the PDB-issued mmCIF file. That way, our files have a table that provides a correspondence between the 1-to-N numbering, the UniProt numbering, and the residue numbering of the original mmCIF file obtained from the PDB (Fig 3).

For the space-delimited legacy PDB format files, if the line starts with ("ATOM"), ("TER"), ("ANISOU") or ("SIGUIJ") the program gets columns 22:26, 27, 17:20, 21 as residue number (AuthResNum), insertion code (InsCode), residue name (AuthResName), and ChainID respectively. For special lines "REMARK 465" records which list missing residues, columns 20:26, 27, 15:18, 19 are obtained as the residue number, insertion code, residue name, and ChainID correspondingly. The two data frames are merged and if the residue does not have a UniProt residue number then it gets default_PDB_num (default_PDB_num = 5000 + SeqResNum). In order to prevent possible numbering collisions, PDBrenum calculates available numbers in the range from 1 to 9999 and reassigns them to non-polymeric compounds in reverse order. After PDBrenum makes the replacement dictionary out of two python strings where the

```
#
loop_
_pdbx_poly_seq_scheme.asym_id
_pdbx_poly_seq_scheme.entity_id
_pdbx_poly_seq_scheme.seq_id
_pdbx_poly_seq_scheme.mon_id
_pdbx_poly_seq_scheme.ndb_seq_num
_pdbx_poly_seq_scheme.pdb_seq_num
_pdbx_poly_seq_scheme.auth_seq_num
_pdbx_poly_seq_scheme.pdb_mon_id
_pdbx_poly_seq_scheme.auth_mon_id
_pdbx_poly_seq_scheme.pdb_strand_id
_pdbx_poly_seq_scheme.pdb_ins_code
_pdbx_poly_seq_scheme.hetero
A 1 1    THR 1   18  18   THR THR A . n
A 1 2    PRO 2   19  19   PRO PRO A . n
A 1 3    LYS 3   20  20   LYS LYS A . n
A 1 4    PRO 4   21  21   PRO PRO A . n
A 1 5    LYS 5   22  22   LYS LYS A . n
A 1 6    ILE 6   23  23   ILE ILE A . n
A 1 7    VAL 7   24  24   VAL VAL A . n
A 1 8    LEU 8   25  25   LEU LEU A . n
A 1 9    VAL 9   26  26   VAL VAL A . n
A 1 10   GLY 10  27  27   GLY GLY A . n
A 1 11   SER 11  28  28   SER SER A . n
A 1 12   GLY 12  29  29   GLY GLY A . n
A 1 13   MET 13  30  30   MET MET A . n
A 1 14   ILE 14  31  31   ILE ILE A . n
A 1 15   GLY 15  32  32   GLY GLY A . n
A 1 16   GLY 16  33  33   GLY GLY A . n
A 1 17   VAL 17  35  35   VAL VAL A . n
A 1 18   MET 18  36  36   MET MET A . n
A 1 19   ALA 19  37  37   ALA ALA A . n
A 1 20   THR 20  38  38   THR THR A . n
A 1 21   LEU 21  39  39   LEU LEU A . n
A 1 22   ILE 22  40  40   ILE ILE A . n
```

```
#
loop_
_pdbx_poly_seq_scheme.asym_id
_pdbx_poly_seq_scheme.entity_id
_pdbx_poly_seq_scheme.seq_id
_pdbx_poly_seq_scheme.mon_id
_pdbx_poly_seq_scheme.ndb_seq_num
_pdbx_poly_seq_scheme.pdb_seq_num
_pdbx_poly_seq_scheme.auth_seq_num
_pdbx_poly_seq_scheme.pdb_mon_id
_pdbx_poly_seq_scheme.auth_mon_id
_pdbx_poly_seq_scheme.pdb_strand_id
_pdbx_poly_seq_scheme.pdb_ins_code
_pdbx_poly_seq_scheme.hetero
A 1 1    THR 1   2    18   THR THR A . n
A 1 2    PRO 2   3    19   PRO PRO A . n
A 1 3    LYS 3   4    20   LYS LYS A . n
A 1 4    PRO 4   5    21   PRO PRO A . n
A 1 5    LYS 5   6    22   LYS LYS A . n
A 1 6    ILE 6   7    23   ILE ILE A . n
A 1 7    VAL 7   8    24   VAL VAL A . n
A 1 8    LEU 8   9    25   LEU LEU A . n
A 1 9    VAL 9   10   26   VAL VAL A . n
A 1 10   GLY 10  11   27   GLY GLY A . n
A 1 11   SER 11  12   28   SER SER A . n
A 1 12   GLY 12  13   29   GLY GLY A . n
A 1 13   MET 13  14   30   MET MET A . n
A 1 14   ILE 14  15   31   ILE ILE A . n
A 1 15   GLY 15  16   32   GLY GLY A . n
A 1 16   GLY 16  17   33   GLY GLY A . n
A 1 17   VAL 17  18   35   VAL VAL A . n
A 1 18   MET 18  19   36   MET MET A . n
A 1 19   ALA 19  20   37   ALA ALA A . n
A 1 20   THR 20  21   38   THR THR A . n
A 1 21   LEU 21  22   39   LEU LEU A . n
A 1 22   ILE 22  23   40   ILE ILE A . n
```

**Fig 3. Renumbering of the pdbx_poly_seq_scheme table from 2aa3 processed by PDBrenum.** Left: the original file from the PDB. Right: the renumbered file from PDBrenum. The original author numbering is given in column 6 of the table on the left (18, 19, 20, etc.), which is replaced with the UniProt numbering (entry Q9PRK9) for this chain (2,3,4, etc) in column 6 of the table on the right. The original author numbering has been placed in column 7 of the table on the right (i.e., in the auth_seq_num position).

keys are AuthResNum (4 Char) + InsCode (1 Char) + ResName (3 Char) + ChainID (1 Char) and the values are UniProtResNum + InsCode + ResName + ChainID, where the values have the same number of characters. Finally, PDBrenum processes each line of the PDB file, replacing keys with values.

The value offset for non-UniProt residues, default_PDB_num (5,000), can be reset (with "--set_default_mmCIF_num" flag) and default_mmCIF_num (50,000) (with "--set_default_PDB_num" flag) as you wish but we recommend it to be big enough so it will not be the same as any other numbers but not bigger then 9000 for PDB format because it might go over the 4-character limit of 9999).

There are some chains in the PDB that are chimeras containing sequence from two or more UniProt entries. In cases where there is no collision of residue numbering, then the sequences are numbered according to the UniProt sequences in the SIFTS file. In cases when there is a collision, the longest sequence is taken as the default. The only exception to this is for a small number of UniProt sequences that are commonly used as crystallization chaperones, and are therefore not the target of main interest. These include UniProt codes GFP_AEQVI, GCN4_YEAST, C562_ECOLX, ENLYS_BPT4, MALE_ECOLI.

Mutations in a UniProt sequence in a PDB entry are annotated in SIFTS files, and the mutated residue type and residue number are retained within the output mmCIF or PDB files.

All the files that were processed by PDBrenum get the name tag "_renum" (e.g. 2aa3_renum.cif.gz) and we insert REMARKs in them.

## REMARK for mmCIF

```
loop_
_database_PDB_remark.id 1
_database_PDB_remark.text
;File processed by PDBrenum: http://dunbrack.fccc.edu/PDBrenum
Author sequence numbering is replaced with UniProt numbering according
to
alignment by SIFTS (https://www.ebi.ac.uk/pdbe/docs/sifts/).
Only chains with UniProt sequences in SIFTS are renumbered.
Residues in UniProt chains without UniProt residue numbers in SIFTS
(e.g., sequence tags) are given residue numbers 50000+label_seq_id
(where label_seq_id is the 1-to-N residue numbering of each chain.
Ligands are numbered 50000+their residue number in the original file.
The _poly_seq_scheme table contains a correspondence between the
1-to-N sequence (seq_id), the new numbering based on UniProt
(pdb_seq_num =
auth_seq_id in the _atom_site records), and the author numbering
in the original mmCIF file from the PDB (auth_seq_num).
;
#
```

## REMARK for PDB legacy

```
REMARK 0 File processed by PDBrenum: http://dunbrack.fccc.edu/
PDBrenum
REMARK 0 Author sequence numbering is replaced with UniProt numbering
REMARK 0 according to alignment by SIFTS
REMARK 0 (https://www.ebi.ac.uk/pdbe/docs/sifts/).
REMARK 0 Only chains with UniProt sequences in SIFTS are renumbered.
REMARK 0 Residues in UniProt chains without UniProt residue numbers in
SIFTS
REMARK 0 (e.g., sequence tags) are given residue numbers 5000
+label_seq_id
REMARK 0 (where label_seq_id is the 1-to-N residue numbering of each
chain.
REMARK 0 Ligands are numbered 5000+their residue number in the
original
REMARK 0 file. The _poly_seq_scheme table contains a correspondence
between
REMARK 0 the 1-to-N sequence (seq_id), the new numbering based on
UniProt
REMARK 0 (pdb_seq_num = auth_seq_id in the _atom_site records), and
REMARK 0 the author numbering in the original mmCIF file from the PDB
REMARK 0 (auth_seq_num).
```

## Setting up PDBrenum

As a prerequisites, anaconda should be installed https://docs.anaconda.com/anaconda/install/.
The following will set up a conda environment for running PDBrenum locally:

```
(base) $ git clone https://github.com/Faezov/PDBrenum.git
(base) $ cd PDBrenum
(base) $ conda create -n PDBrenum python = 3.6 numpy = 1.17 pan-
das = 0.25.1 biopython = 1.76 tqdm = 4.36.1 ipython = 7.8.0
requests = 2.25.1 lxml = 4.6.2
(base) $ conda activate PDBrenum
```

If the Python packages listed above are already installed (or similar versions of them), then PDBrenum may work outside of a conda environment. The user can try it that way, and if no errors are returned, then the program works.

## Running PDBrenum

Help can be obtained with this command:
```
(PDBrenum) $ python3 PDBrenum.py -h
```
The user can provide PDBids directly as a list of arguments (-rfla
- -renumber_from_list_of_arguments):
```
(PDBrenum) $ python3 PDBrenum.py -rfla 1d5t 1bxw 2vl3 5e6h -mmCIF
(PDBrenum) $ python3 PDBrenum.py -rfla 1d5t 1bxw 2vl3 5e6h -PDB
(PDBrenum) $ python3 PDBrenum.py -rfla 1d5t 1bxw 2vl3 5e6h
 -mmCIF_assembly
(PDBrenum) $ python3 PDBrenum.py -rfla 1d5t 1bxw 2vl3 5e6h
 -PDB_assembly
```
or put PDBids in a text file (comma, space, tab, or newline delimited) (-rftf
- -renumber_from_text_file):
```
(PDBrenum) $ python3 PDBrenum.py -rftf input.txt -mmCIF
(PDBrenum) $ python3 PDBrenum.py -rftf input.txt -PDB
(PDBrenum) $ python3 PDBrenum.py -rftf input.txt -mmCIF_assembly
(PDBrenum) $ python3 PDBrenum.py -rftf input.txt -PDB_assembly
```
The user can renumber the entire PDB in a given format (by default in mmCIF if no format is provided):
```
(PDBrenum) $ python3 PDBrenum.py -redb -mmCIF
(PDBrenum) $ python3 PDBrenum.py -redb -PDB
(PDBrenum) $ python3 PDBrenum.py -redb -mmCIF_assembly
(PDBrenum) $ python3 PDBrenum.py -redb -PDB_assembly
```
Note that sometimes on Windows installations, the biopython module might be installed incorrectly by conda and it will cause a module error in python. To resolve this problem simply run:
```
(PDBrenum) $ pip install biopython = = 1.76
```
PDBrenum was heavily tested on all PDB structure files in both formats and on all popular operating systems (Linux, Mac and Windows).

PDBrenum uses multiprocessing by default using all available cores, but the user can set a limit to the number of processors by providing number to -nproc flag.

The user can also change where input output files will go by using these self-explanatory flags (with absolute paths):
```
"-sipm", "--set_default_input_path_to_mmCIF"
"-sipma", "--set_default_input_path_to_mmCIF_assembly"
"-sipp", "--set_default_input_path_to_PDB"
"-sippa", "--set_default_input_path_to_PDB_assembly"
"-sips", "--set_default_input_path_to_SIFTS"
"-sopm", "--set_default_output_path_to_mmCIF"
"-sopma", "--set_default_output_path_to_mmCIF_assembly"
"-sopp", "--set_default_output_path_to_PDB"
"-soppa", "--set_default_output_path_to_PDB_assembly"
```
By default, files go here:
```
default_input_path_to_mmCIF = current_directory + "/mmCIF"
default_input_path_to_mmCIF_assembly = current_directory +
"/mmCIF_assembly"
default_input_path_to_PDB = current_directory + "/PDB"
default_input_path_to_PDB_assembly = current_directory +
"/PDB_assembly"
```

```
SP PDB_id chain_PDB   chain_auth   UniProt   SwissProt     uni_len   chain_len   renum  5k_or_50k
+  2aa3    A           A            Q4PRK9    Q4PRK9_PLAVI      315         316     315          1
+  2aa3    B           B            Q4PRK9    Q4PRK9_PLAVI      315         315     315          0
+  2aa3    C           C            Q4PRK9    Q4PRK9_PLAVI      315         318     315          3
+  2aa3    D           D            Q4PRK9    Q4PRK9_PLAVI      315         315     315          0
```

**Fig 4. Log file of PDBrenum on PDB entry 2aa3.** In this logfile, "SP" means "special case" and denotes which chains were handled in specific ways. It is "+" for cases where there is no clash in UniProt numbers. It is "*" for the case where a chain has more than one UniProt in SIFTS; the UniProt that represents the largest portion of the chain is taken unless it is in the exception list of UniProts often used as crystallization chaperones [GFP_AEQVI, GCN4_YEAST, C562_ECOLX, ENLYS_BPT4, MALE_ECOLI]. PDB_id is the 4-character long PDB identifier; chain_PDB is the chain identifier given by the PDB in mmCIF files (label_asym_id); chain_auth is a chain identifier given by authors of the structure (auth_asym_id); UniProt is the 6-character long UniProt identifier (e.g. Q4PRK9); SwissProt is the human-readable UniProt identifier (e.g., P53_HUMAN); uni_len is the number of residues in the chain represented in the UniProt sequence; chain_len represent the length of the chain sequence; renum represents total quantity of residues that were renumbered according to UniProt; 5k_or_50k represents quantity of residues that were renumbered by adding 5000 to 1-to-N numbering for the PDB-legacy format or adding 50000 to the 1-to-N numbering for the mmCIF format.

```
default_input_path_to_SIFTS = current_directory + "/SIFTS"
default_output_path_to_mmCIF = current_directory + "/output_mmCIF"
default_output_path_to_mmCIF_assembly = current_directory +
"/output_mmCIF_assembly"
default_output_path_to_PDB = current_directory + "/output_PDB"
default_output_path_to_PDB_assembly = current_directory +
"/output_PDB_assembly"
```
Also, by default all files gzipped if you want have them unzipped please use "-offz" or "--set_to_off_mode_gzip"

## Result and discussion

PDBrenum returns a log file and renumbered structure files, shown in Figs 4 and 5 respectively for PDB entry 2aa3. In Fig 5, green arrows point to the columns which were changed.

We have created a webserver (http://dunbrack.fccc.edu/PDBrenum) that will take a list of PDB entry codes or a list of UniProt identifiers (as Accession IDs or SwissProt IDs) and with one click of the mouse, will return a zip file with the requested mmCIF or legacy-PDB format files of the asymmetric units and/or biological assemblies (Fig 6). The output files can also be accessed programmatically via direct http links:

```
http://dunbrack.fccc.edu/PDBrenum/output_PDB/2aa3_renum.pdb.gz
http://dunbrack.fccc.edu/PDBrenum/output_mmCIF/2aa3_renum.cif.gz
http://dunbrack.fccc.edu/PDBrenum/output_PDB_assembly/2aa3_renum.
pdb1.gz
http://dunbrack.fccc.edu/PDBrenum/output_mmCIF_assembly/
2aa3-assembly-1_renum.cif.gz
```

On April 26, 2020, the PDB had 176,507 structures. When processed with PDBrenum:

1. 3,659 structures (2.1%) do not have corresponding SIFTS files (mostly DNA or RNA-only files)

2. 5,919 structures (3.4%) have SIFTS files but do not have any UniProt data in SIFTS (mostly antibodies)

3. 75,359 structures (42.7%) were not changed because all of the proteins have UniProt numbering

4. 23,448 structures (13.3%) were changed due only to the presence of sequence tags or other residues with no UniProt numbers

**Fig 5. Renumbering PDB entry 2aa3.** Screenshot of files 2aa3 before (left) and after (right) PDBrenum: "_atom_site" (coordinate section) (A) and "struct_conf" (B). Green arrows pointed to the columns which were changed.

5. 49,422 structures (28.0%) were changed due to differences between UniProt and author residue numbering only

6. 18,700 structures (10.6%) were changed due to both UniProt/author numbering differences and sequence tags or other non-UniProt residues.

The large percentage of entries (38.6%) that contain proteins that are not numbered according to a standard for each protein (UniProt) demonstrates the need for the approach enabled by PDBrenum.

We feel it is very important that PDBrenum provides renumbered files for both mmCIF and legacy-PDB format. mmCIF is the current standard for PDB files, and provides significantly more information than the original PDB format developed in the 1970s. However, many programs still exist that take only the legacy PDB format, and so it is still worthwhile to provide these files. Finally, we also feel it is important to provide the biological assembly files. More than half of crystal structures in the PDB have annotated assemblies that are different from the asymmetric unit [20]. While RCSB does not provide these files in mmCIF format at this time, they are available from PDBe. Even when the assembly consists of multiple copies of the asymmetric unit, every chain in the assembly files from PDBe has its own unique chainID in the auth_asym_id fields, and so can be processed like any PDB file for further analysis. As an example of the utility of this function in PDBrenum, in Fig 7 we show the result of downloading the renumbered biological assembly structures of UniProt BMP2_HUMAN.

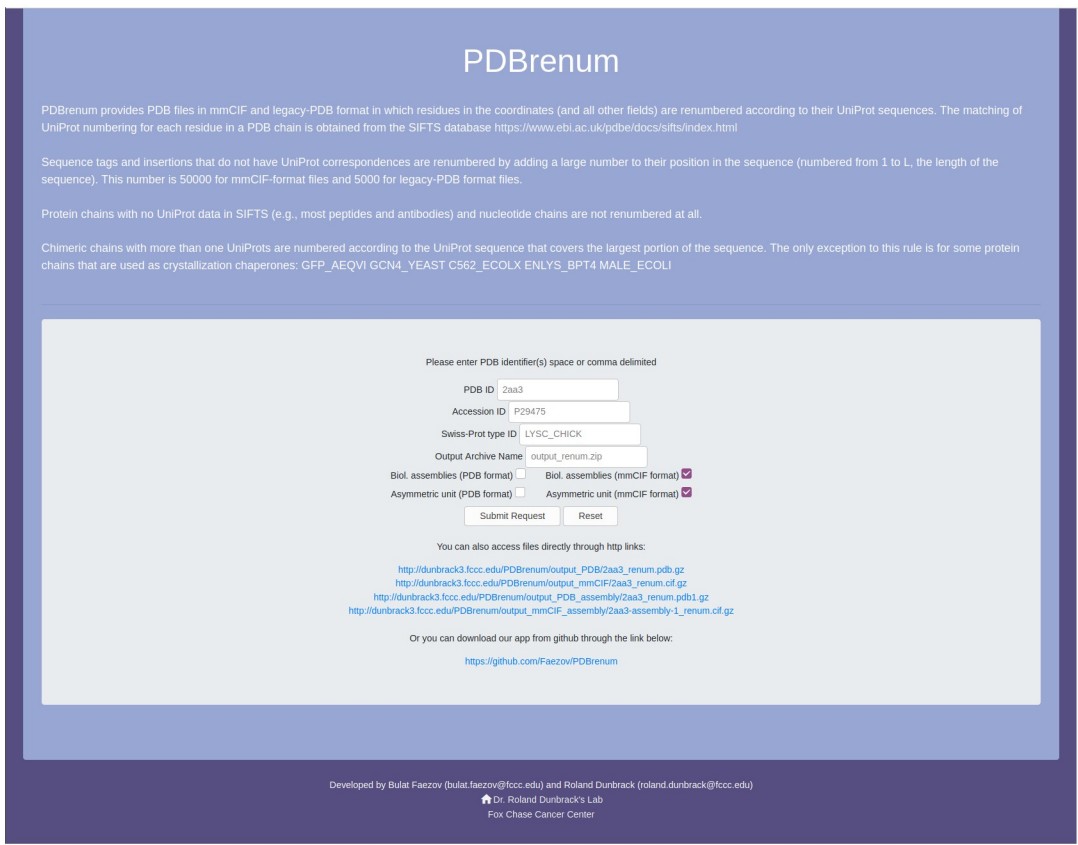

**Fig 6. Screenshot of the PDBrenum server.** The server takes in a list of PDB entry codes (comma, space, tab, or newline separated) or a list of UniProt (e.g. P38398) or SwissProt (e.g., BRCA1_HUMAN) accession codes. The user can choose whether to obtain mmCIF and/or PDB-format files, and whether to obtain asymmetric units and/or biological assemblies with check boxes. If more than one file is requested, a zip file is returned, and the name of this file can be specified.

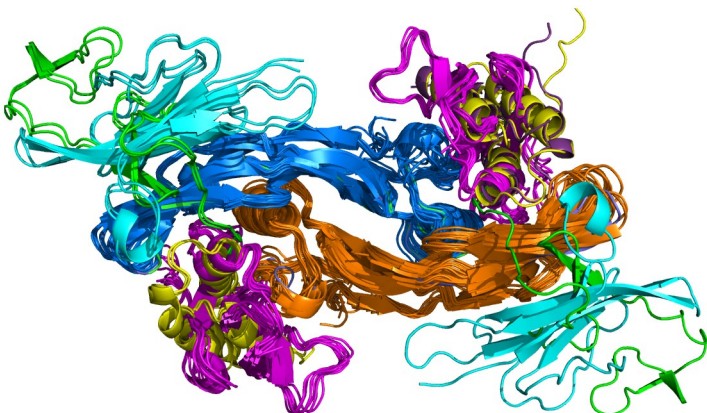

**Fig 7. Biological assemblies of human bone morphogenetic protein 2 (BMP2_HUMAN downloaded with PDBrenum.** BMP2 is a homodimer (orange and blue) that binds Type I (magenta) and Type II receptors (cyan), RGM domain family members (yellow), and von Willebrand factor C-terminal domains (green).

## Acknowledgments

We thank Vivek Modi and Mitchell Parker for testing the PDBrenum program.

## Author Contributions

**Conceptualization:** Roland L. Dunbrack, Jr.

**Data curation:** Bulat Faezov.

**Funding acquisition:** Roland L. Dunbrack, Jr.

**Investigation:** Bulat Faezov.

**Methodology:** Bulat Faezov, Roland L. Dunbrack, Jr.

**Software:** Bulat Faezov.

**Supervision:** Roland L. Dunbrack, Jr.

**Validation:** Bulat Faezov, Roland L. Dunbrack, Jr.

**Visualization:** Bulat Faezov.

**Writing – original draft:** Bulat Faezov.

**Writing – review & editing:** Bulat Faezov, Roland L. Dunbrack, Jr.

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
