## [Decision Letter · Decision Letter 0]

28 May 2021

PONE-D-21-14953

PDBrenum: a webserver and program providing Protein Data Bank files renumbered according to their UniProt sequences

PLOS ONE

Dear Dr. Dunbrack Jr.,

Thank you for submitting your manuscript to PLOS ONE. After careful consideration, we feel that it has merit but does not fully meet PLOS ONE’s publication criteria as it currently stands. Therefore, we invite you to submit a revised version of the manuscript that addresses the points raised during the review process.

We look forward to receiving your revised manuscript.

Kind regards,

Yang Zhang

Academic Editor

PLOS ONE

Journal Requirements:

Reviewers' comments:

Reviewer's Responses to Questions

**Comments to the Author**

1. Is the manuscript technically sound, and do the data support the conclusions?

Reviewer #1: Yes

Reviewer #2: Yes

2. Has the statistical analysis been performed appropriately and rigorously? 

Reviewer #1: N/A

Reviewer #2: Yes

3. Have the authors made all data underlying the findings in their manuscript fully available?

Reviewer #1: Yes

Reviewer #2: Yes

4. Is the manuscript presented in an intelligible fashion and written in standard English?

Reviewer #1: Yes

Reviewer #2: Yes

5. Review Comments to the Author

Reviewer #1: The paper describes PDBrenum, a python tool and webserver to renumber residues according to Uniprot in mmCIF or PDB files. This paper address a significant problem, as the authors show that ~38% of PDB contains proteins that are not numbered according to Uniprot. The code is available on github and the webserver is online and working.

I have some minor concerns to make the manuscript better.

- Figures 1, 5 and 6 in the pdf have a low resolution and are hard to read.

- In figure 4, there is a mismatch between the picture and its description. Example: "comp_uni" and "human_uni" do not appear in the picture.

- What happens to mutated residues? Are they kept in the file or replaced by the ones of the wild type protein in Uniprot? Is the numbering scheme for them corrected? Maybe the authors could comment about it in the manuscript.

Reviewer #2: The authors provide a renumbering tool for PDB files, which can be used on-line or as a standalone program.

The tool will renumber following UniProt numbering, hence numbering over multiple PDB structures of the same protein will be consistent. This will prove to be or become a most valuable tool.

I have two comments:

- after cloning the git, conda is used to set up the proper environment. This is not necessary if the environment is already suited for use of PDBrenum (correct python, etc). conda is not always distributed as an easy to install

package on linux distributions. A note should be added that for use of standalone PDBrenum, and in absence of conda, the following packages with versions blabla should be installed.

- the images were of extremely low resolution. so low that I could hardly read figure 1, let alone the figure with the green lines indicating was had been changed.

6. PLOS authors have the option to publish the peer review history of their article (what does this mean?). If published, this will include your full peer review and any attached files.

Reviewer #1: **Yes: **Ariane Nunes Alves

Reviewer #2: No

---

## [Author Response · Author response to Decision Letter 0]

3 Jun 2021

Dear Dr. Zhang,

We thank the editor and the reviewers for their helpful comments. We have revised the manuscript in response to the reviewers’ concerns as detailed below.

Best wishes,

Bulat Faezov

Roland Dunbrack

(Responses to reviewers’ comments are prefixed with “**”).

Reviewer #1: The paper describes PDBrenum, a python tool and webserver to renumber residues according to Uniprot in mmCIF or PDB files. This paper address a significant problem, as the authors show that ~38% of PDB contains proteins that are not numbered according to Uniprot. The code is available on github and the webserver is online and working.

I have some minor concerns to make the manuscript better.

- Figures 1, 5 and 6 in the pdf have a low resolution and are hard to read.

** In the PDF, PLOS ONE provided a much lower resolution of the figures than we submitted and at the top of each page with a figure there is a link to the submitted tif files that the reviewer can click to get the higher resolution figures. In any case, we have created even higher resolution versions of the screenshot figures by using a higher-resolution monitor. Please use the links at the top of each figure page to evaluate the resolution.

- In figure 4, there is a mismatch between the picture and its description. Example: "comp_uni" and "human_uni" do not appear in the picture.

** Thank you for pointing this out. The caption was written for a previous version of the log file and has now been corrected.

- What happens to mutated residues? Are they kept in the file or replaced by the ones of the wild type protein in Uniprot? Is the numbering scheme for them corrected? Maybe the authors could comment about it in the manuscript.

** Mutated residues are kept with the residue types in the original PDB or mmCIF files. SIFTS annotates mutations that do not match the UniProt residue type. The following sentence was added to the paper:

** Mutations in a UniProt sequence in a PDB entry are annotated in SIFTS files, and the mutated residue type and residue number are retained within the output mmCIF or PDB files

Reviewer #2: The authors provide a renumbering tool for PDB files, which can be used on-line or as a standalone program.

The tool will renumber following UniProt numbering, hence numbering over multiple PDB structures of the same protein will be consistent. This will prove to be or become a most valuable tool.

I have two comments:

- after cloning the git, conda is used to set up the proper environment. This is not necessary if the environment is already suited for use of PDBrenum (correct python, etc). conda is not always distributed as an easy to install package on linux distributions. A note should be added that for use of standalone PDBrenum, and in absence of conda, the following packages with versions blabla should be installed.

** We have added a statement to the README.md file that tells the user that PDBrenum may work outside of a conda environment if biopython, pandas, numpy, tqdm, ipython, lxml, and requests are all installed. The user can try PDBrenum and if no error is returned, then it works with the versions of these packages that the user already has. The user can also install the versions of these packages listed in the README file one by one. Also, wget and curl commands can be used to download files programmatically from the webserver if there are any problems with the python version. We added the following to the paper also:

** If the Python packages listed above are already installed (or similar versions of them), then PDBrenum may work outside of a conda environment. The user can try it that way, and if no errors are returned, then the program works.

- the images were of extremely low resolution. so low that I could hardly read figure 1, let alone the figure with the green lines indicating was had been changed.

** In the PDF, PLOS ONE provided a much lower resolution of the figures than we submitted and at the top of each page with a figure there is a link to the submitted tif files that the reviewer can click to get the higher resolution figures. In any case, we have created even higher resolution versions of the screenshot figures by using a higher-resolution monitor. Please use the links at the top of each figure page to evaluate the resolution.

---

## [Editor Report · Decision Letter 1]

7 Jun 2021

PDBrenum: a webserver and program providing Protein Data Bank files renumbered according to their UniProt sequences

PONE-D-21-14953R1

Dear Dr. Dunbrack Jr.,

We’re pleased to inform you that your manuscript has been judged scientifically suitable for publication and will be formally accepted for publication once it meets all outstanding technical requirements.

Kind regards,

Yang Zhang

Academic Editor

PLOS ONE
---

## [Editor Report · Acceptance letter]

25 Jun 2021

PONE-D-21-14953R1 

PDBrenum: a webserver and program providing Protein Data Bank files renumbered according to their UniProt sequences 

Dear Dr. Dunbrack Jr.:

I'm pleased to inform you that your manuscript has been deemed suitable for publication in PLOS ONE. Congratulations! Your manuscript is now with our production department. 

Kind regards, 

on behalf of

Dr. Yang Zhang 

Academic Editor

PLOS ONE